# CXCL1: A new diagnostic biomarker for human tuberculosis discovered using Diversity Outbred mice

Deniz Koyuncu[1], Muhammad Khalid Khan Niazi[2], Thomas Tavolara[2],
Claudia Abeijon[3], Melanie L. Ginese[3], Yanghui Liao[4], Carolyn Mark[5], Aubrey Specht[3],
Adam C. Gower[6], Blanca I. Restrepo[7], Daniel M. Gatti[8], Igor Kramnik[9],
Metin Gurcan[2], Bülent Yener[10], Gillian Beamer[3]*

1 Rensselaer Polytechnic Institute, Department of Electrical, Computer, and Systems Engineering, Troy, New York, United States of America, 2 Wake Forest School of Medicine, Bowman Gray Center for Medical Education, Winston-Salem, North Carolina, United States of America, 3 Tufts University, Cummings School of Veterinary Medicine, North Grafton, Massachusetts, United States of America, 4 Nanyang Technological University, Singapore, 5 Kansas State University, College of Veterinary Medicine, Manhattan, Kansas, United States of America, 6 Boston University Clinical and Translational Science Institute, Boston, Massachusetts, United States of America, 7 The University of Texas Health Science Center at Houston School of Public Health in Brownsville, Texas, United States of America, 8 The Jackson Laboratory, Bar Harbor, Maine, United States of America, 9 Boston University, National Emerging Infectious Diseases Laboratories, Boston, Massachusetts, United States of America, 10 Rensselaer Polytechnic Institute, Department of Computer Science, Troy, New York, United States of America

* Gillian.Beamer@tufts.edu

**Data Availability Statement:** The gene expression data is uploaded to Gene Expression Omnibus (GEO) and with Series ID GSE179417 and to be released upon publication. The biomarker data is

## Abstract

More humans have died of tuberculosis (TB) than any other infectious disease and millions still die each year. Experts advocate for blood-based, serum protein biomarkers to help diagnose TB, which afflicts millions of people in high-burden countries. However, the protein biomarker pipeline is small. Here, we used the Diversity Outbred (DO) mouse population to address this gap, identifying five protein biomarker candidates. One protein biomarker, serum CXCL1, met the World Health Organization's Targeted Product Profile for a triage test to diagnose active TB from latent *M.tb* infection (LTBI), non-TB lung disease, and normal sera in HIV-negative, adults from South Africa and Vietnam. To find the biomarker candidates, we quantified seven immune cytokines and four inflammatory proteins corresponding to highly expressed genes unique to progressor DO mice. Next, we applied statistical and machine learning methods to the data, i.e., 11 proteins in lungs from 453 infected and 29 non-infected mice. After searching all combinations of five algorithms and 239 protein subsets, validating, and testing the findings on independent data, two combinations accurately diagnosed progressor DO mice: Logistic Regression using MMP8; and Gradient Tree Boosting using a panel of 4: CXCL1, CXCL2, TNF, IL-10. Of those five protein biomarker candidates, two (MMP8 and CXCL1) were crucial for classifying DO mice; were above the limit of detection in most human serum samples; and had not been widely assessed for diagnostic performance in humans before. In patient sera, CXCL1 exceeded the triage diagnostic test criteria (>90% sensitivity; >70% specificity), while MMP8 did not. Using Area Under the Curve analyses, CXCL1 averaged 94.5% sensitivity and 88.8% specificity for active pulmonary TB (ATB) vs LTBI; 90.9% sensitivity and 71.4% specificity for

uploaded to http://www.dsrc.rpi.edu/?page=databank under the headline Dataset accompanying "CXCL1: A new diagnostic biomarker for human tuberculosis discovered using Diversity Outbred mice."

**Funding:** Support was provided by NIH UL1-TR001430 (BUMSR-Boston University Microarray and Sequencing Resource) NIH R21 AI115038 (GB); NIH R01 HL145411 (GB); the American Lung Association Biomedical Research Grant RG-349504 (GB); the Boehringer Ingelheim Veterinary Scholars Research Program V340PR0455 (CM); and the Summer Research Program sponsored by the Cummings School of Veterinary Medicine at Tufts University (AS). The funders had no role in study design, data collection and analysis, decision to publish, or preparation of the manuscript.

**Competing interests:** The authors have declared that no competing interests exist.

ATB vs non-TB; and 100.0% sensitivity and 98.4% specificity for ATB vs normal sera. Our findings overall show that the DO mouse population can discover diagnostic-quality, serum protein biomarkers of human TB.

## Author summary

More humans die of tuberculosis (TB) than any other infectious disease, yet diagnostic tools remain limited. Here, we used the Diversity Outbred mouse population to discover candidate protein biomarkers of human TB. By applying statistical methods and machine learning to multidimensional data, we identified CXCL1 and MMP8 as the two most promising protein biomarker candidates. When evaluated in samples from human patients, CXCL1 achieved the World Health Organization's targeted profile for a triage diagnostic test, discriminating active TB from important clinical differential diagnoses: latent *Mtb* infection and non-TB lung disease in HIV-negative adults. Overall, our studies show how a translationally relevant animal population model can accelerate TB biomarker discovery, validation, and testing for humans.

## Introduction

Tuberculosis (TB) remains a global health crisis. The disease is diagnosed in 8–10 million new patients each year and has a stagnant annual death rate of 1–1.5 million patients each year. This is nearly 5000 deaths per day, comparable to the average daily death rate due to COVID-19 [1]. Pulmonary TB accounts for 70–80% of all TB cases, is the contagious form of TB, and has a 40–70% case fatality rate if untreated [2–5]. For many decades, inbred laboratory mice provided valuable insight into host resistance to *Mycobacterium tuberculosis* (*Mtb*), by careful experimentation to confirm effects of single cell types and single genes in context of a fixed genetic background. The field is now undergoing a shift by (i) including genetically heterogenous animal models to identify factors that control *Mtb*-induced lung damage in immune competent hosts; and (ii) using new means to identify biomarkers that meet the World Health Organization's (WHO) Target Product Profiles (TPPs) for diagnostic tests, i.e. >90% sensitivity and >70% specificity for a triage diagnostic test; and ≥65% sensitivity and ≥98% specificity for a detection diagnostic test [6–9]. Here, we use the Diversity Outbred (DO) mouse population to help inform human TB biomarker investigation because its genetic diversity rivals human genetic diversity [10], and the phenotype responses to *Mtb* infection better model human TB [11–14]. A fraction of the DO population is highly susceptible and inflammatory lung disease progresses early and rapidly with morbidity and mortality within 60 days of *Mtb* aerosol infection. These progressors develop lung granuloma necrosis, neutrophilic inflammation, and fibrin thrombosis [12,13,15]. These human-like disease features do not develop in C57BL/6 inbred mice, and rarely develop in other inbred strains [16,17].

Here, we infected hundreds of DO mice with a low dose of aerosolized *Mtb* Erdman and examined a unique 119-gene expression signature and inflammatory and immunological mediators to find protein biomarker candidates. After exhaustively searching statistical models and machine learning algorithms to identify optimal sets of biomarkers in DO mouse lungs, two combinations performed better than the WHO TPPs when applied to independent DO mouse population data: Logistic Regression with matrix metalloproteinase 8 (MMP8) and Gradient Tree Boosting [18] with a panel of four biomarkers (CXCL1, CXCL2, tumor necrosis

Factor (TNF), and interleukin-10 (IL-10)). This identified five protein biomarker candidates to test in human sera. From these five, we down selected to MMP8 and CXCL1 because these two were essential in their respective algorithms; had not been thoroughly investigated previously; and both were consistently above the limits of detection in sera from human patients. The others (TNF, CXCL2, and IL-10) were not consistently above the limit of detection and were not pursued in depth in human sera. We also tested S100A8 (calgranulin A) as a candidate protein biomarker in human sera to extend prior work by Gopal *et al* [11].

When tested in human sera, only CXCL1 met the WHO's TPP triage diagnostic test criteria, successfully discriminating active pulmonary TB (ATB) from LTBI; from non-TB lung disease; and from normal sera. CXCL1 had 94.5% (95% confidence intervals (CI), 85.1–98.5%) sensitivity and 88.8% (95% CI, 80.5–93.8%) specificity for ATB vs LTBI; and 90.9% (95% CI, 80.4–96.1%) sensitivity and 71.4% (95% CI, 57.6–82.2%) specificity for ATB vs non-TB. CXCL1 also maintained a 100.0% (95% CI, 93.5–100.0%) sensitivity and 98.4% (95% CI, 91.7–99.9%) specificity for ATB vs normal sera. MMP8 met TPP triage criteria for only one comparison: ATB vs normal sera with 100% sensitivity and specificity. MMP8 did not meet WHO TPP triage test criteria for ATB vs LTBI (76.4% (95% CI, 63.7–85.6%) sensitivity and 60.7% (95% CI, 50.3–70.2%) specificity) and ATB vs non-TB (70.9% (95% CI, 57.9–81.2%) sensitivity and 59.2% (95% CI, 45.2–71.8%) specificity). The candidate protein biomarker S100A8 did not achieve TPP triage or detection criteria for any comparison.

Overall, using the DO mouse population lung gene expression profiles to guide protein biomarker candidates, coupled with extensive statistical and machine learning analyses, produced a successful diagnostic quality, serum protein, biomarker for ATB in humans. The best performing protein biomarker, CXCL1 had high sensitivity and specificity to diagnose HIV-negative, adults with ATB in high burden countries (South Africa and Vietnam), and discriminate those patients from the important clinical differential diagnoses: LTBI and non-TB lung disease.

## Results

### *Mtb* infection and pulmonary TB in DO mice

When infected with aerosolized *Mtb*, approximately one-third of DO mice rapidly succumb to inflammatory lung disease with high bacterial burden within 60 days [12], recently termed progressors by Ahmed et al in 2020 [19] and previously termed "supersusceptible" by Niazi et al in 2015 [12]. The progressor phenotype is reproducible across sexes, institutions, aerosol infection methods, and strains of *Mtb* [12,13,19]. Progressor DO mice significantly reduce survival of the DO population, compared to age- and sex-matched non-infected DO mice; and compared to *Mtb*-infected C57BL/6J inbred mice (Fig 1A). The survival drop reflects a mortality peak 20–35 days after *Mtb* infection (Fig 1B) and morbidity is detected as early as 4 days after *Mtb* (Fig 1C), when progressors start to lose body condition. The remaining ~70% of *Mtb*-infected DO mice are recently termed *controllers* by Ahmed et al in 2020 [19] and (previously termed "susceptible" and "resistant" by Niazi et al in 2015 [12]. Controllers have no external signs of morbidity, show normal body condition, and appear active and alert for at least 60 days after infection, as do C57BL/6J inbred mice.

### Lung transcriptome of progressor DO mice identifies potential biomarkers candidates

Immune deficiency does not explain early-onset pulmonary TB in progressor DO mice. All DO mice generate *Mtb* antigen specific TH1 immunity; and are protected by *M. bovis* BCG

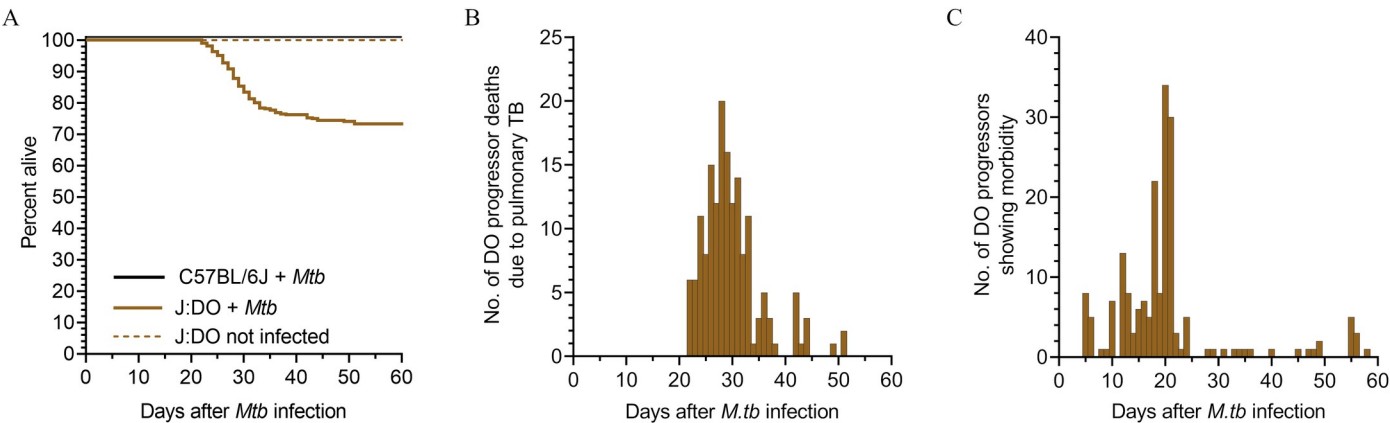

**Fig 1. Survival, deaths due to pulmonary TB, and morbidity onset.** We infected 8-10-week-old, female DO (n = 657) and C57BL/6J (n = 66) mice with ~25 *Mtb* bacilli by inhalation, monitored for health daily, and euthanized the progressor DO mice that developed any one of three IACUC-approved morbidity criteria: body condition score <2; severe lethargy; or respiratory distress (i.e. increased respiratory rate and effort). Within 60 days, 34% of DO mice succumbed to pulmonary TB and required euthanasia due to pulmonary TB (progressors), while 66% survived (controllers). All non-infected DO mice (n = 40) and all *Mtb*-infected C57BL/6J inbred mice survived 60 days without morbidity or mortality ($p < 0.0001$ by Log-rank Mantel-Cox test) (A). When progressor DO succumbed, a mortality wave occurred between 21–52 days and peaked 25–35 days (B). Morbidity in progressor DO mice began as early as 4 days after *Mtb* infection and peaked at 18 days post-infection (C).

vaccination rather than developing BCGosis [12,13,20] which occurs in immune deficient states. DO progressors and controllers also produce IL-17 and *Mtb* specific antibodies indicative of TH17 and TH2 responses, respectively (S3 Fig). Our prior work identified three neutrophil chemokines: CXCL1, CXCL2, CXCL5 that diagnosed progressor DO mice with modest accuracy [12]. To identify additional potential biomarkers, we followed 32 *Mtb*-infected DO mice for 157 days, and profiled total lung RNA for gene expression by microarray. Ten progressor DO mice succumbed to pulmonary TB before 60 days; 11 controllers succumbed to pulmonary TB between 95–157 days; and 11 controllers survived 157 days with no morbidity and were euthanized at 157 days. Five age- and sex-matched non-infected DO controls were also profiled. An analysis of variance revealed that 20,623 of the 25,206 interrogated genes were significantly differentially expressed (FDR $q < 0.05$) across the three groups. Additional filters (fold change > 2 for both the progressor vs non-infected and progressor vs controller comparisons) were then applied to obtain a set of 119 genes that are specifically and highly expressed in the lungs of progressor DO mice (Fig 2). Of those 119 genes, three: *S100a8*, *Mmp8*, and *Cxcl2* appeared in 18 of 31 pathways using Enrichr (S1 Table). As expected, based on lung granuloma features of progressor DO mice [12,14,21], all transcriptomic-identified pathways in the lungs of progressor DO mice converged on acute inflammation: Neutrophil (granulocyte) recruitment and degranulation; positive inflammatory signaling via cytokines and chemokines; and extracellular matrix degradation (S1 Table).

## Immune and inflammatory proteins in lungs of *Mtb*-infected mice

To generate data sets to identify protein biomarker candidates, we quantified S100A8, CXCL2, and MMP8 (i.e., protein products translated from *S100a8*, *Mmp8*, and *Cxcl2* mRNA, which were abundant in lungs of progressor DO mice); a suite of TH1 proinflammatory cytokines required to resist *Mtb* (i.e., Interferon (IFN)-γ, TNF, and IL-12 (p40 and p70); one immune suppressive cytokine (IL-10); and neutrophil chemokines known to contribute to *Mtb*-induced inflammation: CXCL1, CXCL5 [22–24]. We also included Vascular Endothelial Growth Factor because it is one component of an independently validated, promising biomarker panel identified by Ahmad *et al.* for TB diagnosis [7].

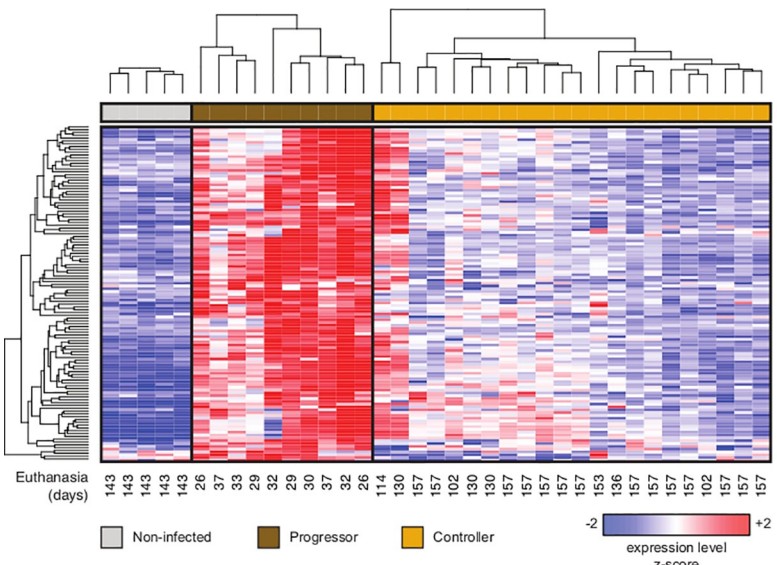

**Fig 2. 119 genes are significantly highly expressed in lungs of progressor DO mice, compared to controller and non-infected DO mice.** Microarray gene expression profiling identified a set of 119 genes whose expression changed significantly across all groups (moderated ANOVA FDR $q < 0.05$) and were upregulated $> 2$-fold in progressor DO mice (n = 10; dark brown) relative to both non-infected DO mice (n = 5; gray) and controller DO mice (n = 22; tan). Rows and columns correspond to genes and individual DO mice, respectively. The number of days from the start of the experiment to the euthanasia of each animal is shown below each column. Hierarchical clustering was performed across all rows (left side of Figure) and was also performed separately within each group (top of Figure). The expression values for each gene were $z$-normalized to a mean of zero and standard deviation of one across all samples in each row; blue, white, and red indicate $z$-scores of $\leq$ -2, 0, and $\geq$ +2, respectively.

We quantified proteins in the lungs of *Mtb*-infected DO mice, C57BL/6J mice, and non-infected age- and sex-matched control DO mice (Fig 3, panels A-K) from 5 independent experimental infections, summarized in Table 1. All proteins except IL-10, IL-12p40, IL-12p70, and VEGF were significantly higher in the lungs of progressor DO mice compared to all other groups.

## Identification of biomarker panels using machine learning and statistical methods

We organized the data from the five different experimental infections of DO mice shown in Table 1 and Fig 3 into discovery and independent cohorts (Fig 4) To identify protein biomarker candidates.

First, we applied best-subset feature selection with 5-fold cross validation to the training data. Data from C57BL/6J mice (n = 42; 10.6% of all data) were included with controller DO mice because C57BL/6J inbred mice survive $> 60$ days without morbidity/mortality (Figs 1 and 3). From the 478 possible classifiers (a classifier refers to an algorithm and its protein biomarker candidates), Logistic Regression with MMP8 (S2 Table) achieved 0.95 AUC, 94.1% sensitivity and 87.4% specificity for classifying progressor DO mice and controller mice (DO and C57BL/6J) on the testing data held back from the discovery cohort (Fig 4). To avoid over-fitting, we validated performance of Logistic Regression with MMP8 on the testing portion of the discovery cohort, achieving 0.96 AUC with 94.4% sensitivity and 87.3% specificity. On data from the independent cohort, the classifier achieved 0.987 AUC (Fig 5A), 78.3% sensitivity, 100% specificity, 100% positive predictive value (PPV) and 94.9% negative predictive value (NPV) (S1 Fig). Although its sensitivity was below the TPP triage test target, Logistic

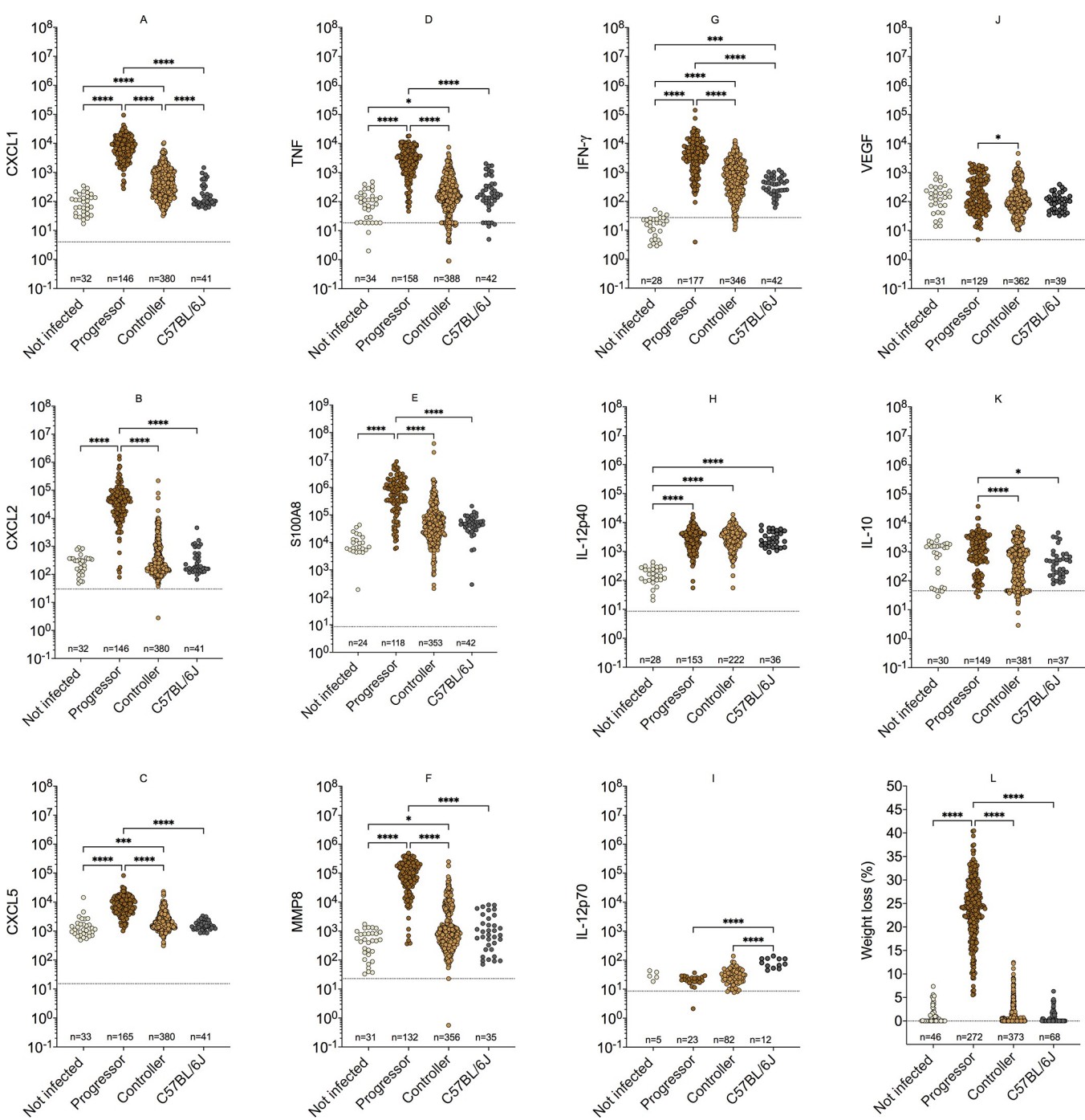

**Fig 3. Lung proteins and weight loss in *Mtb*-infected mice.** We infected 8-10-week-old, female DO (n = 395) and C57BL/6J (n = 42) mice with *Mtb* bacilli by inhalation over 5 independent experimental infections to generate data shown in panels A-K. All progressor DO mice were euthanized by 60 days of *Mtb* infection due to the development of any one of three IACUC-approved early removal criteria: body condition score <2; severe lethargy; or respiratory distress (i.e. increased respiratory rate and effort). The non-infected DO mice, controller DO mice, and C57BL/6J showed no morbidity. We measured lung protein biomarkers using sandwich ELISAs (A-K). In panel (L), weight loss was calculated for each mouse as a percent of its peak body weight from the same 5 independent experiments plus 1 additional experiment for which lung protein data was not available. All data were lognormal distributed and analyzed by Kruskal-Wallis one-way ANOVA with Dunn's multiple comparisons post-tests (*p<0.05; **p<0.01; ***p<0.001; ****p<0.0001). Each dot repsesents 1 mouse.

**Table 1. Mouse class, euthanasia time points (weeks after *Mtb* infection), and numbers of mice used to quantify proteins in lungs.** The first column denotes the class of mice; the second column denotes the euthanasia time point (weeks after *Mtb* infection); and the third column denotes the number of mice euthanized in each week separated by semi columns. There was substantial overlap in euthanasia weeks for the 4 classes. No progressor DO mice survived longer than 8 weeks, so none could be included in the > 14-week timepoint. The table contains information from the same 5 independent experimental infections shown below in Fig 3, panels A-K; and Fig 4.

| Class of mice | Euthanasia time point (weeks after *Mtb* infection) | Number of mice |
|---|---|---|
| Not infected DO | 5; 7; 8; >14 | 15; 5; 9; 5 |
| Progressor DO | 3; 4; 5; 6; 7 | 2; 96; 44; 14; 3 |
| Controller DO | 5; 6; 7; 8; >14 | 80; 61; 116; 66; 45 |
| C57BL/6J | 5; 6; 7; 8 | 10; 6; 6; 20 |

Regression with MMP8 was highly attractive because it used a single biomarker and a linear decision boundary. However, the classifier's specificity across experiments was inconsistent, which led us to identify another classifier that performed with greater consistency.

We sought a classifier capable of 90% sensitivity and 70% specificity in each experimental infection used in discovery. Among the identified classifiers only three (S3 Table) had >80% minimum experiment sensitivity and 70% minimum experiment specificity in the leave-one-experiment-out setting. From the three, we pursued Gradient Tree Boosting with the biomarker panel CXCL1, CXCL2, TNF, and IL-10 because it achieved the highest minimum experiment sensitivity (93.3%), had high experiment-wise sensitivity (97.3%) and overall sensitivity (96.6%).

In the testing portion of the discovery cohort, the four-biomarker panel (CXCL1, CXCL2, TNF, and IL-10) achieved 96.9% and 94.4% sensitivity and specificity, respectively, across all four experiments combined. In the leave-one-experiment-out setting, the panel had 87.5% and 70% minimum experiment sensitivity and specificity, respectively. In the independent cohort, the four-biomarker panel achieved 91.3% sensitivity and 81.7% specificity, satisfying the WHO TPP performance criteria for a triage test. The PPV and NPV of the classifier were 55.3% and 97.4%, respectively (S1 Fig), and its AUC was 0.95 (Fig 5B). We evaluated the classifier further using the 27 age- and sex-matched non-infected DO mice and it achieved 100% specificity. Of the 4 biomarkers in the panel, CXCL1 and CXCL2 had more influence in classification performance than TNF or IL-10, shown by higher variable importance values of 0.62, 0.32, 0.04, and 0.02, respectively.

To determine how each protein biomarker candidate contributes to classifier performance, we measured the average percent difference. MMP8 achieved the highest effect, with an average percent difference of 11.31%, followed by CXCL1 (5.8%) and CXCL2 (4.82%) (S4 Table). Fig 6 demonstrates this effect graphically by showing that biomarker panels lacking CXCL1, CXCL2, or MMP8 have lower AUC values than the biomarker sets that contain at least 1 of these 3.

## Performance of MMP8 and CXCL1 to diagnose human patients with active TB

We pursued MMP8 and CXCL1 as candidate diagnostic biomarkers because neither have been fully investigated in humans and both performed well under different conditions in DO mice. MMP8 is the biomarker used in the first classifier and CXCL1 had the highest variable importance among the panel used in the second classifier. We also quantified and examined S100A8 (calgranulin A) for diagnostic performance in human samples to follow up Gopal *et al* [11]. We obtained sera from HIV-negative adults from South Africa and Vietnam with ATB, LTBI, or non-TB lung disease from the Foundation for Innovative Diagnostics. Pooled serum from healthy individuals served as controls. MMP8, CXCL1, and S100A8 protein levels were

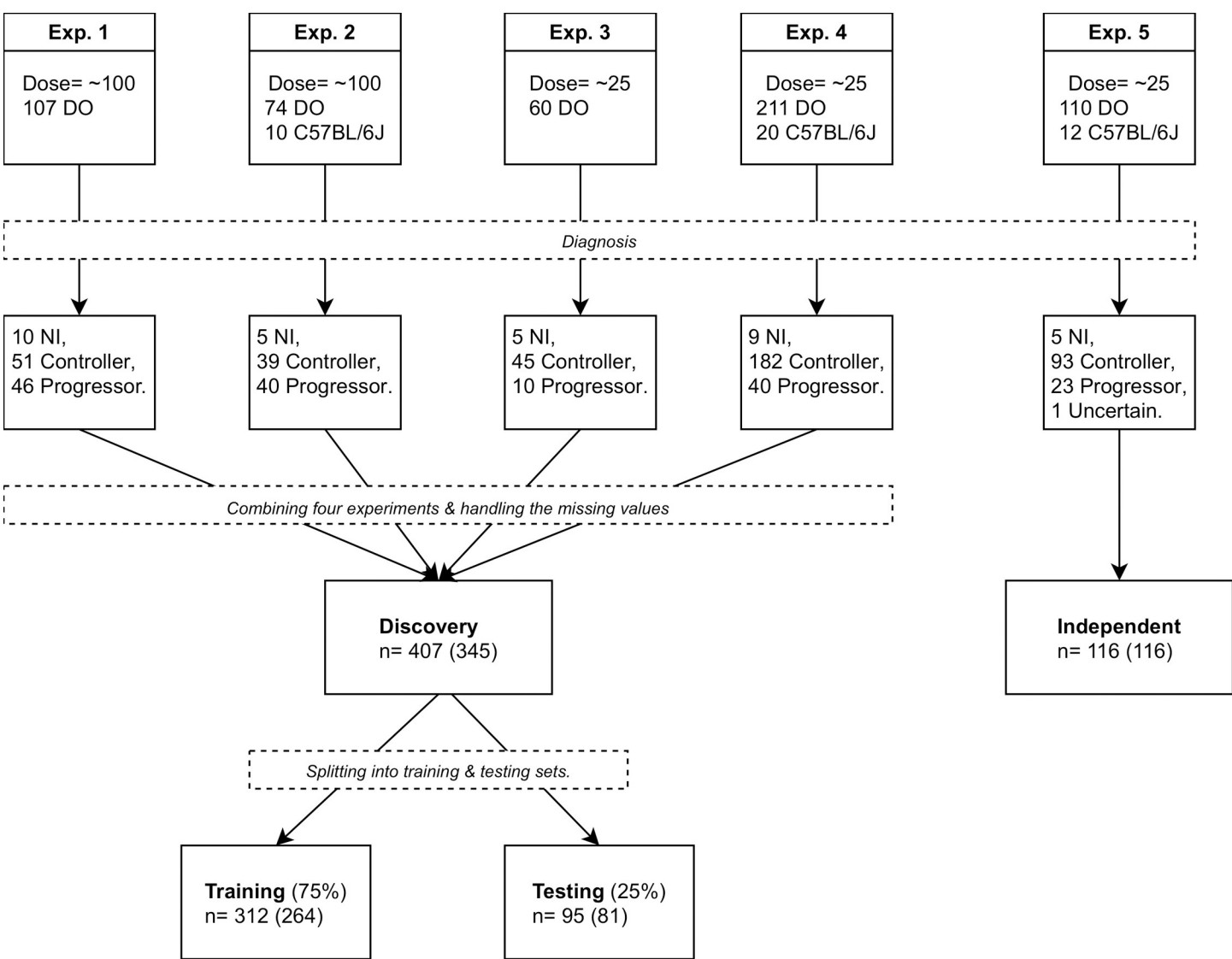

**Fig 4. Flow chart of sample organization and datasets.** The top most five boxes denote the initial *Mtb* dose and the number of DO and C57BL/6J (if used) mice for each of the experiments. The succeeding five boxes report the number of mice in each class. We used the controller and progressor mice from Exp. 1, Exp. 2, Exp. 3 and Exp. 4 in the discovery phase; and controller and progressor mice from Exp. 5 in the independent evaluation. The first number to the right of "n =" denotes the number of samples that do not have any missing values in CXCL5, CXCL2, CXCL1, IFN-γ, TNF, IL-12, IL-10, and their meta data, and the second one within the paranthesis denotes the number of samples that do not have any missing lung biomarkers.

significantly higher in sera from patients with ATB to other groups (Fig 7). MMP8 was above the limit of detection in all samples tested. CXCL1 was above the limit of detection in all serum samples from ATB patients but not the other 3 patient categories; and S100A8 was below the assay limit of detection for many samples in all patient categories.

The following was observed for determining diagnostic performance of MMP8, CXCL1, and S100A8 by AUC analyses, summarized in Table 2. For ATB vs normal sera, MMP8 had perfect AUC (1.00), 100% (95% CI, 93.5–100.0%) sensitivity and 100% (95% CI, 94.4–100.0%) specificity; CXCL1 had 0.999 (95% CI, 0.998–1.00) AUC, 100.0% (95% CI, 93.5–100.0%) sensitivity and 98.4% (95% CI, 91.7–99.9%) specificity; and S100A8 had 0.77 (95% CI, 0.674–0.865) AUC, 72.7% (95% CI, 59.8–82.7%) sensitivity and 60.0% (95% CI, 47.9–71.0%) specificity (S2 Fig). For ATB vs LTBI, MMP8 had 0.774 (95% CI, 0.691–0.857) AUC, 76.4% (95% CI,

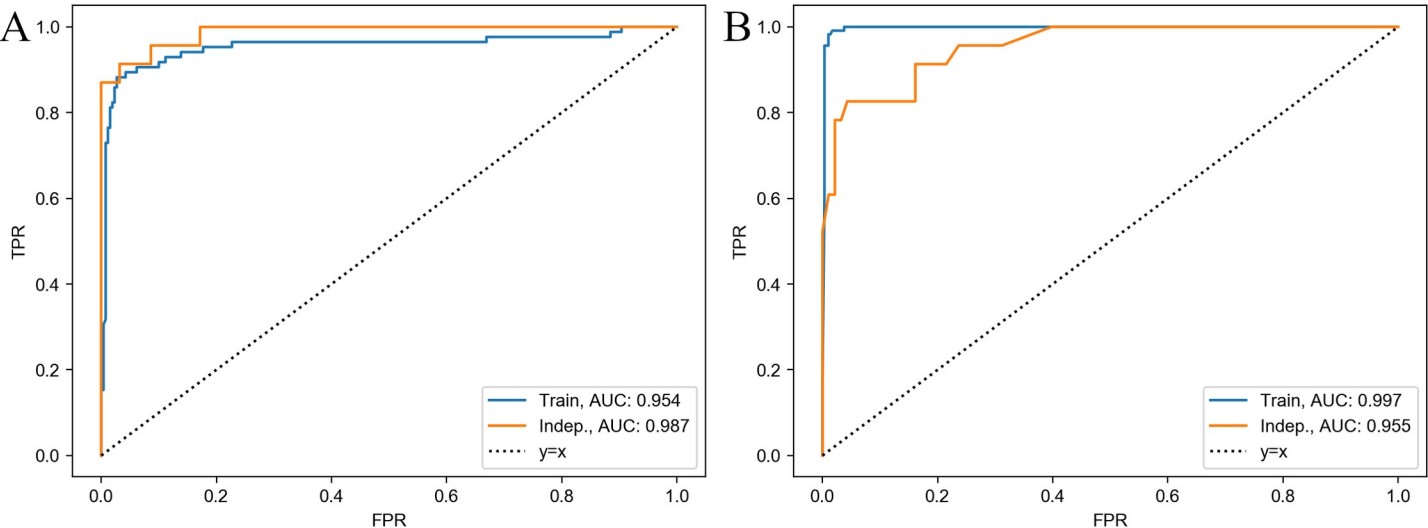

**Fig 5. Receiver Operating Characteristics (ROC) curves of classifier performance to diagnose progressor *Mtb*-infected DO mice.** A) Logistic Regression with MMP8, and B) Gradient Tree Boosting with CXCL1, CXCL2, TNF, and IL-10. Blue and orange curves correspond to the AUC in the discovery cohort (n = 345 for A and n = 407 for B) and independent cohort (n = 116) respectively for both A and B.

63.7–85.6%) sensitivity and 60.7% (95% CI, 50.3–70.2%) specificity; CXCL1 had 0.972 (95% CI, 0.950–0.994) AUC, 94.5% (95% CI, 85.1–98.5%) sensitivity and 88.8% (95% CI, 80.5–93.8%) specificity; and S100A8 had 0.742 (95% CI, 0.653–0.831) AUC, 72.7% (95% CI, 59.8–82.7%) sensitivity and 64.0% (95% CI, 53.7–73.2%) specificity (S2 Fig). For ATB vs non-TB, MMP8 had 0.741 (95% CI, 0.647–0.835) AUC, 70.9% (95% CI, 57.9–81.2%) sensitivity and 59.2% (95% CI, 45.2–71.8%) specificity; CXCL1 had 0.921 (95% CI, 0.871–0.970) AUC, 90.9% (95% CI, 80.4–96.1%) sensitivity and 71.4% (95% CI, 57.6–82.2%) specificity; and S100A8 had 0.764 (95% CI, 0.669–0.858) AUC, 74.5% (95% CI, 61.7–84.2%) sensitivity and 71.4% (95% CI, 57.6–82.2%) specificity (S2 Fig).

## Discussion

The WHO introduced the TPPs in 2014, to unify and promote development efforts for TB diagnostics appealing to end-user requirements for establishing blood-based biomarkers [9]. A

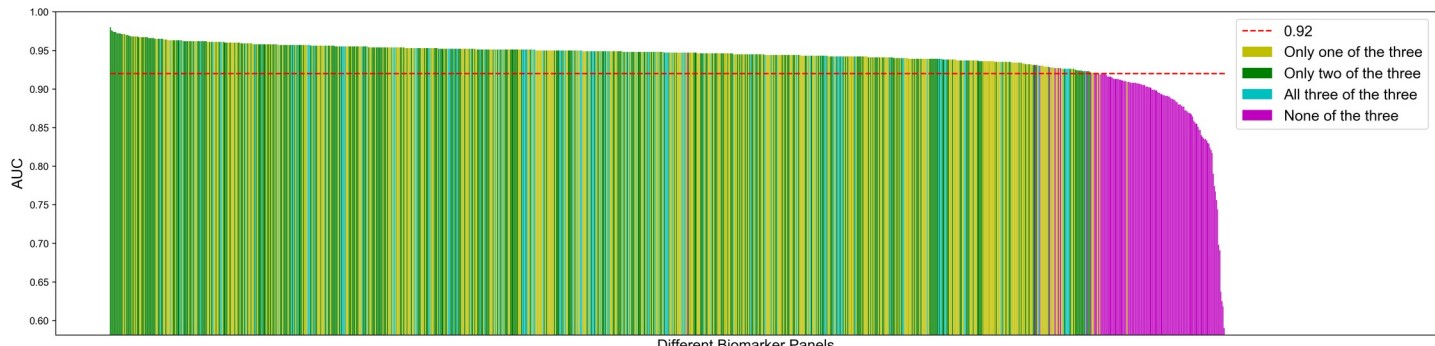

**Fig 6. Classifiers using lung protein biomarker candidates CXCL1, CXCL2, or MMP8 have the highest performance.** Bar chart of 5-fold-cross validation AUC of 1023 different biomarker panels sorted in descending order. Y-axis denotes the AUC and each bar in the x-axis corresponds to a different panel. Yellow, green and teal bars indicate the classifiers using any one of the three lung protein biomarker candidates, any two of the three, and all three respectively. Magenta bars indicate the classifiers that did not include CXCL1, CXCL2 or MMP8. Classifiers that did not include CXCL1, CXCL2, or MMP8 all had AUCs lower than 0.92, the red dashed line.

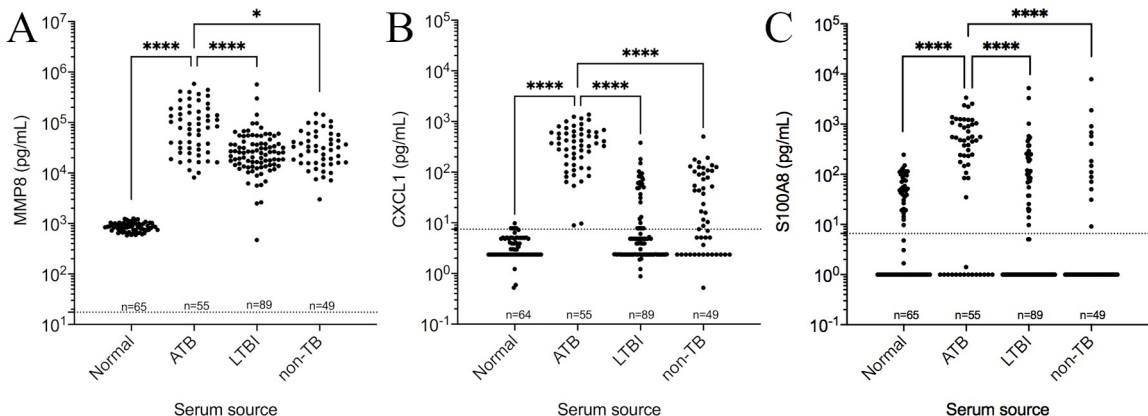

**Fig 7. MMP8, CXCL1 and S100A8 protein levels in human sera from patients with active pulmonary TB (ATB), latent *Mtb* infection (LTBI), non-TB disease and normal individuals.** We tested serum from HIV-negative patients ATB, LTBI, non-TB disease and replicates from pooled normal for MMP8 (A), CXCL1, (B) and S100A8 (C) by ELISA, and analyzed data by Kruskal-Wallis one-way ANOVA with Dunn's multiple comparisons post-tests ($^*<0.05$, $^{****}p<0.0001$). Dashed lines show the limits of detection (LOD): MMP8 LOD = 17.47 pg/mL, CXCL1 LOD = 7.46 pg/mL, and S100A8 LOD = 6.59 pg/mL.

systematic review of TB biomarker papers published between 2010 and 2017, identified 44 panels or single biomarkers that satisfy a TPP criterion, and simultaneously determined that only one panel had a low risk of bias using QUADAS-2 assessments (study design, sampling, negative population, timing, reference standard and blinding) [6]. The panel was ECM1, myoglobulin, HCC1, IL-21, ENA-78, TPA, IL-12(p40) and IL-13 which achieved 100% (95% CI, 83.2–100%) sensitivity and 95% (95% CI, 68.1–99.9%) specificity; however, sample size was relatively small [25]. A more recent study identified a protein biomarker panel consisting of IL-6, IL-8, IL-18, VEGF, and anti-Ag85 which nearly achieved TPP criteria on hundreds of patient samples from geographically distinct regions [7]. Guidance from the systematic review [6] and results from Ahmad *et al* [7] demonstrate larger samples and additional approaches are needed for biomarker discovery, validation, and independent testing.

To address that need, we model human responses to *Mtb* by using the DO mouse population. Like humans, DO mice are highly genetically diverse and show variable responses and outcomes following aerosol infection with virulent *Mtb* [12,13]. In 2015, we classified *Mtb*-infected DO mice as supersusceptible, susceptible, and resistant using 166 *Mtb*-infected DO mice and we generated a biomarker decision tree that diagnosed the classes with about 70%

**Table 2. Serum protein biomarker AUC analyses in human patient sera.** AUC, sensitivity, and specificity values of CXCL1, MMP8, and S100A8 are given for active pulmonary TB (ATB) vs pooled normal, ATB vs latent *Mtb* infection (LTBI) infection, and ATB vs non-TB comparisons. We denoted the 95% confidence intervals in the parenthesis. For each comparison and each column, the highest value is highlighted in bold.

| | Name | AUC | Sensitivity (%) | Specificity (%) |
|---|---|---|---|---|
| ATB vs Normal | CXCL1 | 0.999 (0.998–1.00) | 100.0 (93.5–100.0) | 98.4 (91.7–99.9) |
| | MMP8 | **1.00** | **100.0 (93.5–100.0)** | **100.0 (94.4–100.0)** |
| | S100A8 | 0.77 (0.674–0.865) | 72.7 (59.8–82.7) | 60.0 (47.9–71.0) |
| ATB vs LTBI | CXCL1 | **0.972 (0.950–0.994)** | **94.5 (85.1–98.5)** | **88.8 (80.5–93.8)** |
| | MMP8 | 0.774 (0.691–0.857) | 76.4 (63.7–85.6) | 60.7 (50.3–70.2) |
| | S100A8 | 0.742 (0.653–0.831) | 72.7 (59.8–82.7) | 64.0 (53.7–73.2) |
| ATB vs non-TB | CXCL1 | **0.921 (0.871–0.970)** | **90.9 (80.4–96.1)** | **71.4 (57.6–82.2)** |
| | MMP8 | 0.741 (0.647–0.835) | 70.9 (57.9–81.2) | 59.2 (45.2–71.8) |
| | S100A8 | 0.764 (0.669–0.858) | 74.5 (61.7–84.2) | **71.4 (57.6–82.2)** |

accuracy [12]. In 2020, Ahmed *et al* [19] put forth different terms (progressor and controller) based on lung RNAseq from 29 *Mtb*-infected DO mice, effectively aligning terms and immune correlates with lung responses in *Mtb*-infected macaques. Here, we have dropped "supersusceptible, susceptible, and resistant" adopted the progressor and controller terminology as well. With > 500 *Mtb*-infected DO mice in our data set, we classify progressors based on survival < 60 days (Fig 1), who develop lung disease characterized by neutrophilic influx [12], granuloma necrosis [15], and convergent pathways of acute inflammation (S1 Table). We have also established means to diagnose progressors with high accuracy (91.50 ± 4.68%) by several other methods including a novel "imaging biomarker" generated via artificial intelligence, which performs comparably to expert veterinary pathologists (> 94.95%) using H&E stained-lung sections [14]. We very recently established a lung 5-gene expression signature (*serpina3n*, *ifitm6*, *serpina3m*, *ms4a8a* and *cxcr2*) that performed well on cross-validation (n = 77 DO mice) and external testing (n = 33 DO mice), achieving sensitivity and specificities of 88% and > 93% in both conditions [21].

Here, we show how the DO mouse population can be used to find translationally relevant protein biomarkers for human ATB. We identified two classifiers that discriminate progressor DO mice from controllers (and from non-infected DO mice): (i) Logistic Regression with MMP8 and (ii) Gradient Tree Boosting with a panel of four biomarkers (CXCL1, CXCL2, TNF, and IL-10). The first classifier had an advantage of being a single biomarker and a simple regression model, and MMP8 satisfied the minimal requirements of a WHO TPP detection test. However, the classifier's performance varied across experiments attributed to the different criterion used in feature selection. The second classifier performed consistently across independent experiments and achieved the WHO triage test specifications with the operating points selected in the discovery cohort.

In our studies, two proteins from the lungs of DO mice (MMP8, and CXCL1) were the most promising biomarker candidates. When tested in sera from HIV-negative adults from countries with high TB burden (South Africa and Vietnam), both MMP8 and CXCL1 were significantly increased in serum from patients with ATB compared to patients with LTBI, to patients with non-TB lung disease, and to pooled sera from normal individuals. However, only CXCL1 had sensitivity and specificity higher than the WHO recommendations for a triage test to diagnose human patients with ATB. CXCL1 maintained high AUCs, successfully discriminating ATB from LTBI (0.972) and ATB from non-TB lung disease (0.921). MMP8 did not meet TPP criteria and its AUCs were substantially lower (0.774 for ATB vs LTBI and 0.741 for ATB vs non-TB), because the sample distribution ranges overlapped.

The human gene *CXCL1* was previously identified as part of a neutrophil-driven gene expression signature common in blood samples from ATB patients.[26] However, the proteins MMP8 and CXCL1 have not been thoroughly investigated. TB biomarkers with the highest AUC values reported in humans include C-reactive protein, transferrin, IFN-γ, IP-10, IL-27, and interferon–inducible T Cell Alpha Chemoattractant [27–35]. Only two studies reported MMP8 as a biomarker [36,37] and three studies measured CXCL1 [38–40]. Albuquerque *et al*. [36] reported only the diagnostic performance of a panel containing MMP8 and not its univariate performance. Kathamuthu et al. [37] measured eight different MMP family proteins including MMP8 in plasma samples and reported an AUC for MMP8 as 0.774 for ATB vs healthy controls. Like our results (AUC of 0.774), investigators reported AUC of 0.799 for MMP8 to discriminate ATB vs LTBI. However, an important patient group was not included in their MMP8 analyses: non-TB lung disease. A few other investigators quantified MMP2, MMP3 and MMP9 (proteins functionally related to MMP8) but did not pursue them further as diagnostic biomarker candidates due to missing data [7], or inconsistent statistical differences [27,35]. Three studies measured CXCL1 [38–40] and identified statistically significant

differences in TB patients, but were not analyzed for diagnostic performance due to small sample sizes and different sample types: n = 44 to 88 for plasma [38]; n = 11 to 27 for serum [39]; and n = 11 to 27 and n = 32 to 72 for saliva [39,40].

Prior work suggested the protein multimer S100A8/S100A9 (calprotectin) as a potential biomarker by detecting statistically higher protein levels [11] and *s100a8* and *s100a9* gene expression [19] in DO mice, and in patients with ATB compared to LTBI and to healthy controls [11]. The studies did not report diagnostic performance, however. Here, we identified one of the two components of calprotectin, S100A8 (calgranulin A) and its gene *s100a8* were significantly higher in progressor DO mice compared to controller. In human sera, S100A8 was also significantly higher in ATB compared to non-TB lung disease, LTBI, and pooled sera from normal individuals but many samples were below the limit of detection (Fig 7). When we tested S100A8's performance as a diagnostic serum protein biomarker, it failed to meet the WHO TPP criteria. We did not examine S100A9 (calgranulin B) or the multimer S100A8/A9 for diagnostic performance.

Recently published work by Ahmad *et al* in 2020[19], compared genes differentially expressed in DO mouse and macaque lungs; and blood samples from humans to identify immune correlates of disease and resistance across different species and disparate samples. We examined their publicly available data for the gene names of our 11 candidate protein biomarkers. All 11 of our biomarker candidate genes' expression were significantly increased in one or more comparisons of progressor vs controller; and progressor vs naive in DO mouse and macaque lungs described by Ahmad *et al*. Specifically, *ifng, tnf, cxcl1, vegfa, cxcl2,* and *vegfd* were increased in progressor for all four comparisons; *il10, il12b, s100a8,* and *mmp8* were increased in progressor in 3 of 4 comparisons; *vegfb* and *cxcl5* were increased in 2 out of the 4 comparisons; *Vegfc* was increased in only one of the comparisons. None of our 11 protein biomarker candidates' gene names were amongst Ahmed *et al's* list of the top 10 highly expressed genes, which is not surprising, given that Ahmed *et al* focused on different comparisons, with fewer samples (n = 29), and used a different selection strategy. None of our 11 protein biomarker candidates were amongst the 13-mouse gene orthologs described as aligning with the 16-gene predictive signature from human blood samples (i.e., the 16-gene ACS signature) [19]. Presumably, this reflects host species differences, tissue sample differences, or both. Notably, two of our 11 lung protein biomarker candidates (S100A8 and MMP8) overlap with genes up regulated in human blood in Ahmed *et al*, and none were down regulated.

Our work here discovering, validating, and independently testing TB biomarkers in DO has some limitations. First, we used two separate datasets in the discovery cohort (n = 407 or 345) because of missing data. Ideally, when comparing two different classifiers, we should use identical data sets to estimate out-of-sample performance. However, since mouse lungs are small with limited volumes, not all lung samples were available to measure all biomarkers. This contributed to missing values, and for simplicity, samples with missing values were excluded from training, validation, and testing. Second, the WHO TPPs were intended for non-sputum samples of humans [9] and not lung samples from experimentally infected mice. We believe, however, that benchmarking against the TPPs increases our ability to find potential biomarker candidates, although we may me missing biomarkers that are unique to blood. Our future studies will include non-lung samples from *Mtb*-infected DO mice. Third, the number of human serum samples we used is smaller than the recent biomarker publications such as [7] which has 179 TB patients and 138 patients with non-TB-lung disease. Our aim, however, was to show a proof of concept that DO mice can discover translationally relevant TB biomarkers. We hope that other investigators will reproduce our findings in larger studies of TB patients with different forms of disease, co-morbidities, and patients with non-TB lung disease.

Overall, we show two important findings: First, that DO responses to *Mtb* exceed the ranges of inbred mice, clearly demonstrating genetic control of variable susceptibility to *Mtb*. Their responses mimics human disease and resistance phenotypes, providing the field with a powerful *in vivo* model for discovery. Second, that MMP8 and CXCL1 identified, validated, and independently tested in lung samples from DO mice, are translationally relevant biomarker candidates for human TB that may improve our ability to diagnose TB patients, and are worthy of continued pursuit.

## Materials and methods

### Ethics statement

Tufts Institutional Animal Care and Use Committee (IACUC) approved animal experiments under protocols G2012-53; G2015-33; G2018-33. Tufts Institutional Biosafety Committee approved biohazardous infectious agent work under registrations GRIA04; GRIA10; and GRIA17. De-identified serum samples from humans were purchased from the Foundation for Innovative Diagnostics (FIND, Geneva, Switzerland) and approved for testing by Tufts Institutional Review Board study 11675. Serum human pooled normal samples were purchased from MP Biologicals (Santa Ana, California).

### Mice and *Mtb* infection

Female DO (generations 15,16,21,22,34) and C57BL/6J mice were purchased from The Jackson Laboratory (Bar Harbor, ME) and housed in sterile BSL3 conditions at the New England Regional Biosafety Laboratory (Tufts University, Cummings School of Veterinary Medicine, North Grafton, MA). At 8-10-weeks old, mice were infected with aerosolized *Mtb* strain Erdman bacilli (~100 bacilli in experiments 1 and 2; ~25 bacilli in experiments 3, 4, 5) using a CH Technologies nose-only exposure system. All mice were weighed at least twice per week, and examined daily for lethargy, respiratory distress, and body condition [41]. Mice that met morbidity criteria (body condition score < 2; severe lethargy; or respiratory distress prior to 60 days of infection were euthanized immediately, i.e., progressors [19] (previously termed "supersusceptible" [12]). Mice that controlled infection were termed controllers [19] previously termed "not supersusceptible"[12]. All exposed mice were confirmed to be infected by recovering live *Mtb* bacilli from homogenized lung tissue as described [12,20].

### Lung RNA expression profiling and analysis

One lung lobe from each of 42 DO mice was homogenized in TRIzol™ and stored at -80C until RNA extraction using Pure Link mini-kits (Life Technologies, Carlsbad, CA). 37 lung RNA samples were of sufficient purity to be analyzed at the Boston University Microarray and Sequencing Resource Core Facility (Boston, MA). Mouse Gene 2.0 ST CEL files were normalized to produce gene-level expression values using the implementation of the Robust Multiarray Average (RMA) [42] in the Affy package (version 1.62.0) [43] included in the Bioconductor software suite [44] and an Entrez Gene-specific probeset mapping (17.0.0) from the Molecular and Behavioral Neuroscience Institute (Brainarray) at the University of Michigan [45,46]. Array quality was assessed by computing Relative Log Expression (RLE) and Normalized Unscaled Standard Error (NUSE) using the affyPLM package (version 1.59.0). The CEL files were also normalized using Expression Console (build 1.4.1.46) and the default probesets defined by Affymetrix to assess array quality using the AUC metric. Moderated t tests and ANOVAs were performed using the limma package (version 3.39.19) (i.e., creating simple linear models with lmFit, followed by empirical Bayesian adjustment with eBayes). Correction

for multiple hypothesis testing was accomplished using the Benjamini-Hochberg false discovery rate (FDR) [47]. Human homologs of mouse genes were identified using HomoloGene (version 68) [48]. All microarray analyses were performed using the R environment for statistical computing (version 3.6.0). Enrichr (https://amp.pharm.mssm.edu/Enrichr) was used to determine the overrepresentation of Gene Ontology (GO) biological processes (version 2018) within an input set on official mouse gene symbols.

## Quantification of protein biomarkers in DO mouse lungs & human samples

Two lung lobes from each mouse were homogenized in 2mL of phosphate buffered saline and stored at -80C until use. Homogenized lung samples were serially diluted and tested for CXCL5, CXCL2, CXCL1, TNF, MMP8, S100A8, IFN-γ, IL12p40, IL-12p70 heterodimer, IL-10, and VEGF by sandwich ELISA using antibody pairs and standards from R&D Systems (Minneapolis, MN), Invitrogen (Carlsbad, CA), eBioscience (San Diego, CA), or BD Biosciences (San Jose, CA, USA), per kit instructions. Serum samples obtained from FIND were HIV-negative adults. Patients were diagnosed using FIND's criteria. Briefly, patients with ATB were diagnosed by clinical signs, imaging, and confirmed with positive sputum microscopy or culture for *Mtb*. Patients diagnosed with LTBI were immunologically reactive to *Mtb* antigens and lacked clinical, radiographical, and microbiological evidence of ATB. Patients with non-TB lung disease presented with clinical signs of coughing (resembling ATB) but had TB ruled out. For additional details see S6 Table. MMP8, CXCL1, and S100A8 were quantified by ELISA using antibody pairs and standards from R&D Systems (Minneapolis, MN), per kit instructions.

## Statistical analyses

Survival, weight loss, and ELISA data were analyzed and graphed in GraphPad Prism 8.4.2 with significance set at $p < 0.05$ and adjusted for multiple comparisons. Survival curves were analyzed using Log-rank (Mantel-Cox) test. For lung biomarkers and weight loss, data was analyzed for normal or lognormal distributions prior to Kruskal-Wallis one-way ANOVA and Dunn's post-tests, *p<0.05; **p<0.01; ***p<0.001, ****p<0.0001. DO mouse studies for biomarker discovery were based on sample sizes recommended for human biomarker studies [6]. Studies using human sera were pilot studies to generate proof-of-concept results. All available human serum samples were used in ELISA testing. AUC analyses used GraphPad Prism 8.4.3 and reported 95% confidence intervals. If multiple operating points satisfied >90% sensitivity and >70% specificity, we selected the one with the higher Youden's index. If none satisfy the criterion, we selected the operating point closest to the region >90% sensitivity and >70% specificity.

## Machine learning algorithm development, implementation, and validation

**Discovery cohort.** The lung protein measurements from 4 experimental infections with 107, 84, 60, and 231 samples respectively are combined into a dataset with 482 mice in total. Then, non-infected mice are removed, resulting in 453 mice. To address missing data, we considered two scenarios. In the first, 407 mice which do not have any missing values in the following seven biomarkers CXCL5, CXCL2, CXCL1, IFN-γ, TNF, IL-12, and IL-10 are selected. In the second scenario, 345 mice that do not have any missing lung biomarker values are selected. During feature selection, if any of the three potential biomarker candidates MMP8, VEGF, S100A8 are included, then the second set is used. If none of them are included, then the first set is used. The samples are split 75% for classifier selection which includes the

selection of the classification algorithm, its hyper-parameters, and the best subset of biomarkers; and 25% for estimating the unbiased performance of the selection stratified by class (progressor and controller) and experiment number.

**Leave-one-experiment-out setting.**   In the leave-one-experiment-out setting, three of the four experiments in the discovery cohort are combined and used for training, and the remaining one is used as the validation set, resulting in four different train/validation pairs. When the training portion of the discovery cohort is used, the training and validation set sizes of the four pairs are: 253,59; 267,45; 269,43;147,165, respectively. When the testing portion of the discovery cohort is used, the training and validation set sizes of the four pairs are 253,17; 267,13; 269,12;147,53, respectively. If any of the biomarkers MMP8, VEGF, S100A8 are included in the subset selection, then only the samples with complete measurements for all ten biomarkers are used.

**Independent cohort.**   In the independent cohort, there are 122 samples (93 controller, 23 progressor, and 6 not infected). No samples had missing values.

**Experiment-wise metrics.**   First, the sensitivity (specificity) in each experiment is calculated and then averaged. Minimum experiment sensitivity (specificity) is calculated by obtaining the sensitivity (specificity) in each of the experiments and then taking the minimum. In the leave-one-experiment-out setting, experiment-wise sensitivity (specificity) is defined as the average of the four sensitivity (specificity) values. Similarly, minimum experiment sensitivity (specificity) is the minimum of the four sensitivity (specificity) values. To calculate the AUC, an experiment-wise Receiver Operating Characteristics (ROC) curve is drawn where the y-axis corresponds to sensitivity and the x-axis to the specificity and then the area under that curve is calculated.

**Preprocessing.**   Each protein biomarker candidate is standardized by subtracting the sample mean and dividing it by the uncorrected sample standard deviation. During training, only training samples are used to estimate the population parameters. For the first approach during testing, both the samples of the training and the testing sets are used to estimate the population parameters to standardize the testing samples. This holds for testing on Discovery Cohort and Independent Cohort. For the second approach, the population parameters estimated in training are used to standardize the test set.

**Classification algorithm, hyper-parameter, and feature selection.**   We used two different variations of best-subset selection. Using the first approach, we have identified Logistic Regression with MMP8 and using the second approach we identified Gradient Tree Boosting with CXCL1, CXCL2, TNF, and IL-10. During both approaches, for each subset of features in the search space, all classification algorithms and their hyper-parameters are searched. For each subset and each classification algorithm, hyper-parameters with the highest AUC are selected for the first approach, and for the second approach, the ones with the highest experiment-wise AUC are selected. To evaluate the out-of-sample performance, we used 5-fold Cross-Validation (CV) and 100-fold CV for the first and the second approaches, respectively.

*Subsets searched*: The feature search space has all combinations of CXCL5, CXCL2, CXCL, IFN-γ, TNF, IL-12, IL-10, and all combinations of CXCL2, CXCL1, IL-10, IL-12, MMP8, VEGF, S100A8, resulting in 239 different subsets.

**Classification algorithms searched.**   We compared the performance of linear classification algorithms: Support Vector Machine (SVM), Logistic Regression with L1 regularization; and non-linear classification algorithms: SVM with Radial Basis Function, Random Forest, and Gradient Tree Boosting through 5-fold-CV in the training portion of the discovery cohort. The performance of linear and non-linear classification algorithms was similar in their respective categories, so we selected one from each category. The selected two algorithms were Logistic Regression with L1 regularization and Gradient Tree Boosting. Scikit-learn [49] is used for

the implementation of the algorithms. The number of samples in each class is unbalanced; therefore, each sample is re-weighted by its inverse class proportions in the loss functions of both algorithms.

**Hyper-parameters searched.** For Gradient Boosting Tree, logloss is used and as hyper-parameters learning rate (0.001, 0.01), number of trees (1,3,5), and max depth of a tree (1,3,5) are searched for the first approach. For the second approach (0.01, 0.3, 0.5,1.) are searched for the learning rate instead. For the Logistic Regression with L1 regularization, the weight of the L1 loss is selected among (0.01,0.0316, 0.1, 0.316, 0.1) for both approaches. As the result of the search, for the biomarker panel of CXL1, CXCL2, TBF and IL10, learning rate 0.1, number of trees 3, and max depth of a tree 5 are selected. For the Logistic Regression with MMP8 all hyper-parameters, we observed that the weight of L1 loss did not change the AUC for a classifier with a single feature.

**Retraining after classifier selection.** After the classifier selection is complete, the selected classifier with fixed hyper-parameters and selected features is re-trained on all training portion of Discovery Cohort, then evaluated on the testing portion of it. Similarly, the selected classifier is trained on all the Discovery Cohort before it is evaluated on the independent cohort.

**Operating point selection.** We selected the prediction threshold as a hyper-parameter and selected it through K-fold-Cross Validation (CV). In the first approach, 5-fold-CV is used and the threshold that achieves the highest sensitivity while maintaining at least 70% specificity is selected. In the second approach, to reduce the error resulting from selecting a single threshold for K-classifiers, we used a higher number of folds where during the classifier selection 100-fold-CV is used and 150-fold-CV is used when the selected classifier is retrained using both training and the testing datasets. The operating point that maximizes the experiment-wise sensitivity while achieving at least 70% specificity in each of the experiments is selected. If two operating points satisfy the criteria and achieve a similar ($<0.1\%$) experiment-wise sensitivity the one with the higher experiment-wise specificity is selected.

## Supporting information

**S1 STARD Checklist. File contains Standards for Reporting Diagnostic accuracy studies (STARD) checklist.**
(PDF)

**S1 Fig. Confusion matrices for classifying between progressor and controller mice in the independent cohort.** A) Logistic Regression with MMP8 and B) Gradient Tree Boosting with CXCL1, CXCL2, TNF, and IL-10. Row labels and column labels indicate the correct and the predicted labels, respectively.
(TIF)

**S2 Fig. Confusion matrices for the biomarkers in human sera.** The rows in the overall figure correspond to CXCL1, MMP8, and S100A8, respectively. The first column denotes the results for ATB vs pooled normal samples, the second denotes ATB vs LTBI, and the last one denotes ATB vs non-TB. For each confusion matrix row labels and column labels indicate the correct and the predicted labels, respectively.
(PNG)

**S3 Fig.** Additional lung proteins that we measured are shown (A-D). All data were lognormal distributed and analyzed by Kruskal-Wallis one-way ANOVA with Dunn's multiple comparisons post-tests ($^*$p$<0.05$; $^{***}$p$<0.001$). Each dot represents 1 mouse. Dashed lines show the limits of detection (LOD): For A-D LODs are 18.08 pg/mL, 89.27 pg/mL, 24.80 pg/mL, and

84.12 pg/mL, respectively.
(TIF)

**S1 Table. Pulmonary TB in progressor DO mice reflects acute inflammation, neutrophil recruitment and activation, and extracellular matrix degradation.** Enrichr identified the following gene ontology pathways using a set of 119 genes that were highly expressed in the lungs of progressor DO mice compared to non-infected DO mice and to controller DO mice. GO terms with adjusted p < 0.01 are shown, along with the human homologs of the genes overlapping with the term. Expressed genes in bold were pursued as diagnostic biomarkers.
(DOCX)

**S2 Table. 5-fold-Cross Validation (CV) results of the initial selection.** The first three rows achieved the highest AUC and the remaining rows achieved comparable AUC with a simpler model. "Threshold" denotes the probability threshold selected. The standard deviation of the five folds is denoted with ±.
(DOCX)

**S3 Table. The 100-fold-Cross Validation (CV) performance of the three classifiers that satisfy the criterion for both leave-one-experiment-out and four-experiments-combined settings.** Columns under "Leave-one-exp-out" and "Four-exp-combined" correspond to the performance in leave-one-experiment-out setting and four-experiments-combined setting, respectively. "Spec." denotes specificity and "Sens." denotes sensitivity. Sensitivity (specificity) under the "Exp-Wise" columns indicate experiment-wise sensitivity (specificity) and sensitivity (specificity) under the "Min. Exp." columns indicate minimum experiment sensitivity (specificity) and sensitivity (specificity) under the "Overall" columns indicate sensitivity (specificity) (As defined in Materials and methods).
(DOCX)

**S4 Table. Feature rankings of the ten biomarkers.** "Avg." is average and "Diff." is difference. In Materials and Methods, the calculation for average % difference is described. To calculate the average difference, the same method is used but instead of averaging over % difference, it is averaged over difference. Average AUC denotes the average AUC of the biomarker panels that contain the biomarker. Average top 5 denotes the average AUC of the five biomarker panels with highest AUC that contain the biomarker and average bottom 5 denotes the average AUC of the five biomarker panels with the least AUC that contain the biomarker.
(DOCX)

**S5 Table. The AUC values of the ten biomarkers in the discovery cohort for classifying between progressor and controller mice.** 95% confidence intervals are denoted in the parenthesis. AUC values with the confidence intervals were calculated using pROC [50].
(DOCX)

**S6 Table. The patient demographics of the human sera obtained from FIND are shown.** Patients that are missing demographic information are omitted from the results displayed. For rows next to "Country" and "Sex" the number of patients in each category and its percentage is given. At the final row, the median and the IQR of the age is given.
(DOCX)

**S7 Table. The 5-fold-CV AUC values of the 478 classifiers that are searched during the first classifier selection.** Highlighted in yellow are the six selected classifiers whose sensitivity and specificity values are reported in S2 Table. The standard deviation of the five folds is

denoted with ±.
(XLSX)

**S8 Table. The 100-fold-CV performance of the 478 classifiers that are searched during the second classifier selection.** Highlighted in yellow are the three classifiers that satisfied the criterion. Columns under "Leave-one-exp-out" and "Four-exp-combined" correspond to the performance in the leave-one-experiment-out setting and four-experiments-combined setting, respectively. In the leave-one-experiment-out setting, a different experiment is used as the validation set and the columns D-K denote the specificity and sensitivity values on the four validation sets. Columns L and M denote the average of the specificity and sensitivity values in columns D-K, respectively. Columns N and O denote the minimum of the specificity and sensitivity values in columns D-K, respectively. In the four-experiments-combined setting, four experiments are combined into a dataset. Columns P-W denote the specificity and sensitivity values calculated using only the samples from a single experiment. Columns X and Y denote the average of the specificity and sensitivity values in columns P-W, respectively. Columns Z and AA denote the minimum of the specificity and sensitivity values in columns P-W, respectively. Columns AB and AC denote the sensitivity and specificity values computed in the usual way i.e., using all the samples.
(XLSX)

**S1 File. File contains Supplementary Methods, Supplementary Text and Supplementary References.**
(DOCX)

## Acknowledgments

We thank Julie Tzipori, Curtis Rich, Donald Girouard, and Sam Telford III for direction and services in the New England Regional Biosafety Laboratory at Tufts University Cummings School of Veterinary Medicine, North Grafton, MA. Frances Brown, Linda Wrijil, Sarah Ducat, and Gina Scarglia provided histology services at Tufts University's Cummings School of Veterinary Medicine. All microarray protocols were carried out by the Boston University Microarray and Sequencing Resource (BUMSR) core facility, and we thank Eduard Drizik of the BUMSR for initial microarray analyses. Construction of the New England Regional Biosafety Laboratory was made possible by NIH NIAID UC6A1066843. The following reagents were obtained through BEI Resources, NIAID, NIH: ESAT-6, Recombinant Protein Reference Standard, NR-49424; CFP-10, Recombinant Protein Reference Standard, NR49425; Plasmid pMRLB.7 Containing Gene Rv3875 (Protein ESAT-6) from *Mycobacterium tuberculosis*, NR-50170; Plasmid pMRLB.46 Containing Gene Rv3874 (Protein Cfp10) from *Mycobacterium tuberculosis*, NR-13297; *Mycobacterium tuberculosis*, Strain H37Rv, Culture Filtrate Proteins, NR-14825; and *Mycobacterium tuberculosis*, Strain H37Rv, Cell Wall Fraction, NR-14828.

## Author Contributions

**Conceptualization:** Melanie L. Ginese, Igor Kramnik, Metin Gurcan, Bülent Yener, Gillian Beamer.

**Data curation:** Deniz Koyuncu, Adam C. Gower, Blanca I. Restrepo, Daniel M. Gatti, Gillian Beamer.

**Formal analysis:** Deniz Koyuncu, Muhammad Khalid Khan Niazi, Thomas Tavolara, Gillian Beamer.

**Funding acquisition:** Melanie L. Ginese, Igor Kramnik, Metin Gurcan, Bülent Yener, Gillian Beamer.

**Investigation:** Deniz Koyuncu, Claudia Abeijon, Melanie L. Ginese, Yanghui Liao, Carolyn Mark, Aubrey Specht, Blanca I. Restrepo, Metin Gurcan, Gillian Beamer.

**Methodology:** Deniz Koyuncu, Muhammad Khalid Khan Niazi, Thomas Tavolara, Claudia Abeijon, Melanie L. Ginese, Yanghui Liao, Carolyn Mark, Aubrey Specht, Adam C. Gower, Daniel M. Gatti, Metin Gurcan, Gillian Beamer.

**Project administration:** Bülent Yener, Gillian Beamer.

**Software:** Deniz Koyuncu, Thomas Tavolara.

**Supervision:** Bülent Yener, Gillian Beamer.

**Validation:** Deniz Koyuncu, Gillian Beamer.

**Visualization:** Deniz Koyuncu, Adam C. Gower, Gillian Beamer.

**Writing – original draft:** Deniz Koyuncu, Bülent Yener, Gillian Beamer.

**Writing – review & editing:** Deniz Koyuncu, Muhammad Khalid Khan Niazi, Thomas Tavolara, Melanie L. Ginese, Adam C. Gower, Blanca I. Restrepo, Daniel M. Gatti, Igor Kramnik, Metin Gurcan, Bülent Yener, Gillian Beamer.

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
