## [Decision Letter · Decision Letter 0]

15 Apr 2021

Dear Dr. Beamer,

Thank you very much for submitting your manuscript "MMP8 and CXCL1: New translationally relevant tuberculosis biomarkers discovered using Diversity Outbred mice." for consideration at PLOS Pathogens. As with all papers reviewed by the journal, your manuscript was reviewed by members of the editorial board and by several independent reviewers. The reviewers appreciated the attention to an important topic. Based on the reviews, we are likely to accept this manuscript for publication, providing that you modify the manuscript according to the review recommendations.

Sincerely,

Marcel A. Behr

Associate Editor

PLOS Pathogens

JoAnne Flynn

Section Editor

PLOS Pathogens

Kasturi Haldar

Editor-in-Chief

PLOS Pathogens

orcid.org/0000-0001-5065-158X

Michael Malim

Editor-in-Chief

PLOS Pathogens

orcid.org/0000-0002-7699-2064

Reviewer Comments (if any, and for reference):

Reviewer's Responses to Questions

**Part I - Summary**

Reviewer #1: In this manuscript from Koyuncu et. al, they used an impressive number of diversity outbred (DO) mice infected with aerosol Mtb to characterize biomarkers of severe disease. They identify S100a8, Mmp8 and Cxcl2 in supersusceptible DO mice, then validate these and other proteins by ELISA from lung homogenates. They then used computational methods to identify the most promising biomarkers and combination of biomarkers. Finally they used human sera to test MMP8 and CXCL1 in patients with active pulmonary disease, latent Mtb infection and healthy controls and show great discrimination between active TB samples and healthy controls.

While I have a few comments/concerns about the timepoints for the DO mouse analysis, overall I think this is a well-done study. Using animal models like DO mice to better model human TB outcomes is important, and the confirmation of the targets with human sera adds direct translational impact.

Reviewer #2: The purpose of this study was to demonstrate that the Diversity Outbred (DO) panel of mice can be leveraged as an effective systems genetics tool to discover novel diagnostic biomarkers of tuberculosis that can translate to human cohorts. The authors identified two biomarkers of interest, MMP8 and CXCL1, through the machine learning approaches Logistic Regression and Gradient Tree Boosting, respectively. These biomarkers were found to be much more abundant in “supersusceptible” (SS) DO mice when compared against other groups, including uninfected DOs (NI), infected non-SS DO mice (nSS), and infected C57BL/6J (B6) mice. Additionally, when tested in serum samples from HIV-negative patients, these biomarkers were found in much higher concentrations in patients with active TB when compared against healthy controls and patients with latent TB infection.

As a concept, this study actualizes the original vision of the DO panel, utilizing a number of these mice as a phenotypic interrogation of a genetically heterogeneous, mammalian population with SNP-level diversity that exceeds that of the human population. The current paper confirms the author's previous work, that identified CXCL1 as a biomarker of super-susceptibility in mice. The idea that clinically relevant biomarkers could be identified by an outbred population and verified in clinical samples is a compelling experimental direction for research with this particular panel.

Reviewer #3: The manuscript: MMP8 and CXCL1: New Translationally relevant tuberculosis biomarkers discovered using Diversity Outbred mice, by Koyuncu, et all, takes advantage of the genetically driven phenotypic diversity present in the Diversity Outbred (DO) mouse populations to identify lung transcriptional and proteins markers of severe Mtb induced disease. The studies used cohorts of Mtb infected DO mice in combination with machine learning approaches to determine whether protein biomarkers in the lungs of infected mice are associated with severe disease, and this was validated by analysis of an additional DO cohort. They also provide evidence that MMP8 and CXCL1 are associated with human Mtb infection status, suggesting that the results from the DO studies may be relevant to human Mtb. Overall, the study is well done and represents an interesting application of the DO population and provides the field with potentially clinically relevant markers of Mtb disease state. However, there are a few points that should be addressed to improve the manuscript and provide the reader with key information.

**Part II – Major Issues: Key Experiments Required for Acceptance**

Reviewer #1: -It’s unclear to me what timepoint post-infection the microarrays were performed (Figure 2). The authors state that 32 Mtb-infected DO mice were followed for 157 days, but 10 mice developed pulmonary TB before 8 weeks and 22 Mtb-infected DO mice showed no evidence of disease. Does that mean the 10 “supersusceptible” lung RNA samples were taken prior to 8 weeks, but the “Not supersusceptible” lung RNA samples are taken at 157 days (about 22 weeks) post-infection? If the “supersusceptible” samples and “Not supersusceptible” samples were taken at different times post-infection, does this confound the analysis? Is there a way to match timepoints for “supersusceptible” and “Not supersusceptible” mice? Would it be appropriate to include C57Bl/6 mice taken at the same timepoints as the “supersusceptible” mice as controls? The same comment applies for the lung biomarker data in Figure 3. In Figure 3 there are C57Bl/6 controls, but are they also taken at a late timepoint compared to the “SS” mice?

-The authors zeroed in on S100a8, Mmp8 and Cxcl2 from the microarray analysis because they appeared in 18 of 31 pathways, but there’s no mention of fold-change values for these genes or if they were the top hits for differential expression. I feel like there should be more discussion of why these 3 genes were pulled out of 121 identified and if there are other genes that could be analyzed in the future.

Reviewer #2: For general clarity, the machine learning aspects/descriptions could be built on to further explain the terminology, which will help the reader understand many central questions. For example, the definition of the term “classifier” appears at line 134 after three instances of the term in the introduction and is relatively vague. This reviewer was left with questions on the nature of approaches used (Logistic Regression, Gradient Tree Boosting), the reason these two approaches were selected when three of the five approaches mentioned were not (Support Vector Machine, SVM with Radial Basis Function, and Random Forest), the appropriateness of both chosen approaches to the study, how the study design shown in Fig. 4 facilitates these approaches, and the precise meaning and significance of the reported sensitivity, specificity, and AUC statistics. Given that these statistics are reported throughout a good portion of the text, it is additionally unclear why the ROC curves (Fig. S1) and confusion matrices (Fig. S2, which is not referenced in main text, & Fig. S3) are included in the supplement instead of within the main text.

Reviewer #3: 1. Did any of the human TB biomarkers besides IFN-gamma (e.g. C-reactive protein, transferris,...) discussed in lines 230-231 show differential expression in the DO studies. This analysis would likely be limited to the transcriptional analysis (Fig. 2), but it would be worth discussing whether this is the case and the potential impact that this would have on the relevance of the model to human Mtb infection.

**Part III – Minor Issues: Editorial and Data Presentation Modifications**

Reviewer #1: -The definition of “supersusceptible” vs “Not supersusceptible” should be more clear when discussing Figure 1 in the results or the methods. I’m pretty sure those terms are defined by lethality by 8 weeks, but it could be spelled out more clearly.

-Should address the paper from Kathamuthu et. al “Matrix Metalloproteinases and Tissue Inhibitors of Metalloproteinases Are Potential Biomarkers of Pulmonary and Extra-Pulmonary Tuberculosis” Front Immunology (2020) in the discussion.

Reviewer #2: -This study is well-designed to demonstrate the existence of SS DOs, their relatively high Mtb susceptibility indicated by early onset disease and mortality (Fig. 1), and their distinct transcriptional. (Fig. 2) and translational (Fig. 3) profiles in comparison to NI, B6, and nSS mice. However, the truncation of the survival data in Fig. 1 undercuts the human-like spectrum of phenotypes across the DO panel reported on lines 24 and 198. This reviewer is curious if a longer survival curve would better illustrate the two distinct SS & nSS groups.

-Kruskal-Wallis one-way ANOVA, which is used throughout this study, assumes similar group distribution across tested groups, an assumption which does not seem to be met in some of the Fig. 3 subfigures (B, C, K, L). That said, the choice to use non-parametric statistical tests is a good one given that the distribution of these traits in an outbred population will not mirror the canonical normal distribution of phenotypes in an extensively inbred population, such as B6.

-Studies used to train the machine learning model do not all include B6 mice, which begs the question of how (or if) batch normalization was implemented for Exp. 1 & Exp. 3 (as per Fig. 4), especially given that batch variation is noted on line 211.

-With regard to the comment on lines 216-217 stating that this is this first study with DO mice being used for translational biomarker discovery, this does not seem to be entirely accurate. This reviewer would encourage the authors to cite Gopal et al., Am J Respir Crit Care Med (2013), in which S100A8 & S100A9 are examined as biomarkers of TB in DO mice. In addition, Ahmed et al. Sci Transl Med (2020) investigated disease correlates of tuberculosis amongst DO mice, NHP and human cohorts. Further discussion of these other papers would be helpful for readers to get a fuller picture of how the DO are being leveraged for translational discoveries.

-It is unclear why both CXCL1 and CXCL2 are being highlighted in Fig. 5 while the two other members of the four-biomarker panel (TNF, IL-10) are not. It seems like changing the color scheme to only emphasize classifiers with CXCL1 and MMP8 would be more in line with the other results of the study.

-This reviewer found Fig. 6 very difficult to interpret. The asterisks for the statistical tests are placed in such a way that it is difficult to understand what comparisons they represent. Moreover, it is not stated what the error bars on this plot represent. Assuming that the bars represent standard deviation, the variance on the ATB group indicates that this group should not be significantly different from LTBI in either case, yet the difference is declared significant in both. This figure needs to be modified for clarity.

-In the figures where the abbreviations are used, it would be helpful to the reader to list out those abbreviaitons so the figures stand alone (ie. Figure 3, NI, SS, not SS; Figure 4 NI, nSS, SS)

-Some copyedits for the authors’ convenience:

o Line 90: “One third” = “One-third”

o Fig. 1 Legend: “succumned” = “succumbed”

o Fig. 2 Title: “unniquely” = “uniquely”

o Line 192: “specify” = “specificity”

o Line 324: “satisfies” = “satisfy”

o Line 335: “than” = “then”

o Line 597: Double space after “Enrichr”

o Line 749: “leave-one-experiment” = “leave-one-experiment-out”

Reviewer #3: 1. Please provide information on which generation at JAX each DO mouse cohort was derived from, and whether any information is available on kinship relationships between mice from each of the cohorts.

2. It would be helpful if a clearer breakdown were provided on the range of times when the SS mice in Figure 3 were euthanized and how this compares to the other groups (NI, not SS, B6). In general, the results section is somewhat sparse on detail, and the authors should consider providing additional detail through out the results section so that the reader can more readily access important information such as mouse numbers in each experiment, the time of infection for each group, or the number of mice within key groups within Fig. 3. This information can often be found in other parts of the manuscript, but the reader is forced to work for that info in the current version of the manuscript.

PLOS authors have the option to publish the peer review history of their article (what does this mean?). If published, this will include your full peer review and any attached files.

Reviewer #1: No

Reviewer #2: No

Reviewer #3: No

Figure Files:

Data Requirements:

Reproducibility:

References:

---

## [Editor Report · Decision Letter 1]

30 Jun 2021

Dear Dr. Beamer,

We are pleased to inform you that your manuscript 'CXCL1: A new diagnostic biomarker for human tuberculosis discovered using Diversity Outbred mice.' has been provisionally accepted for publication in PLOS Pathogens.

Best regards,

Marcel A. Behr

Associate Editor

PLOS Pathogens

JoAnne Flynn

Section Editor

PLOS Pathogens

Kasturi Haldar

Editor-in-Chief

PLOS Pathogens

orcid.org/0000-0001-5065-158X

Michael Malim

Editor-in-Chief

PLOS Pathogens

orcid.org/0000-0002-7699-2064
---

## [Editor Report · Acceptance letter]

6 Aug 2021

Dear Dr. Beamer,

We are delighted to inform you that your manuscript, "CXCL1: A new diagnostic biomarker for human tuberculosis discovered using Diversity Outbred mice.," has been formally accepted for publication in PLOS Pathogens.

Best regards,

Kasturi Haldar

Editor-in-Chief

PLOS Pathogens

orcid.org/0000-0001-5065-158X

Michael Malim

Editor-in-Chief

PLOS Pathogens

orcid.org/0000-0002-7699-2064